# Feature Mapping for Rice Leaf Defect Detection Based on a Custom Convolutional Architecture

**DOI:** 10.3390/foods11233914

**Published:** 2022-12-04

**Authors:** Muhammad Hussain, Hussain Al-Aqrabi, Muhammad Munawar, Richard Hill

**Affiliations:** 1Department of Computer Science School of Computing and Engineering, University of Huddersfield, Queensgate, Huddersfield HD1 3DH, UK; 2Department of Computer Science, COMSATS University of Islamabad, Park Road, Tarlai Kalan, Islamabad 45550, Pakistan

**Keywords:** leaf smut, bacterial blight, quality inspection, lightweight, deep learning

## Abstract

Rice is a widely consumed food across the world. Whilst the world recovers from COVID-19, food manufacturers are looking to enhance their quality inspection processes for satisfying exportation requirements and providing safety assurance to their clients. Rice cultivation is a significant process, the yield of which can be significantly impacted in an adverse manner due to plant disease. Yet, a large portion of rice cultivation takes place in developing countries with less stringent quality inspection protocols due to various reasons including cost of labor. To address this, we propose the development of lightweight convolutional neural network architecture for the automated detection of rice leaf smut and rice leaf blight. In doing so, this research addresses the issue of data scarcity via a practical variance modeling mechanism (Domain Feature Mapping) and a custom filter development mechanism assisted through a reference protocol for filter suppression.

## 1. Introduction

Rice is considered as one of the most widely consumed foods across the globe, boasting a total consumption rate of 493.13 million metric tons between 2019 and 2020 [1]. For some countries, rice cultivation is a significant source of income generation; for example, Bangladesh achieved its highest GDP (10.73 billion BDT) in 2019 via the agriculture sector, with 50% of the agriculture GDP coming from rice production [2]. Several factors, particularly COVID-19, have accentuated the importance of stringent food quality protocols to ensure exported consumables do not pose a risk to the consumer and the wider community. As a result, countries hoping to gain from rice cultivation via exportation across the globe need to enhance their inspection mechanisms for identifying and removing emerging diseases during the growing season.

Timely and accurate detection of rice plant diseases is fundamental to preserving the quality and quantity of rice production. At present, the de facto approach for quality inspection of rice production is purely manual [3]. This approach is highly vulnerable when it comes to the criterion for quality inspection, i.e., timely and accurate. Human inspection not only takes a significant amount of time but is also costly due to being labor-intensive. Furthermore, the subtle nature of various diseases means domain expertise is required for the task of differentiating between a normal and a defective plant along with the severity of the diseases. With the recent advancements in deep learning and, specifically, computer vision, researchers are actively researching its implementation in quality inspection across various domains including agriculture, production [4], and manufacturing [5]. This paper highlights the present research direction in implementation of deep learning for rice plant disease detection, data scarcity, and architectural design issues and proposes a lightweight custom convolutional neural network for defect detection in rice plants. 

### 1.1. Literature Review

The idea of automating quality inspection processes within various agricultural domains is a well-researched topic expanding across the complete spectrum from conventional image processing [6] utilizing pattern recognition techniques, to machine learning via support vector machines [7]. However, as the focus of our research is on the present state-of-the-art deep learning techniques, we geo-fence our literature review around convolutional-neural-network (CNN)-inspired architectures for rice leaf diseases detection.

Zhou et al. [8] proposed a rice leaf defect detection method based on the fusion of Faster-RCNN and the kmeans algorithm. The framework was initiated via a two-dimensional filter coupled with a multi-level weighted median filter for noise reduction. The authors addressed complex background interference via a segmentation algorithm, Faster 2D-OTSU. The architecture was trained on a dataset consisting of over 30,000 images with the authors reporting impressive accuracies of 96.71%, 97.53%, and 98.26% for rice blast, bacterial blight, and blight, respectively. Although the reported performance is impressive, the large training dataset may be the driving factor for generalization. However, in practical terms, it is not always possible to acquire large amounts of representative data for each type of defect; hence, this paper focuses on representative data generation when starting with a small dataset. Furthermore, the selection of Faster-RCNN [9] is understandable from the perspective of achieving high accuracy due to the two-stage detection process; however, a large amount of data samples is required, and the composition of the internal layers within the Faster-RCNN means it is computationally very demanding. The latter point, i.e., large computational demand, means that the architecture cannot simply be deployed on a standard CPU hardware device for production-based inference. 

Qin et al. [10] presented a cluster-based mechanism coupled with a supervised classification model for leaf lesion segmentation, normalization, and feature extraction. The authors tested their framework on images of alfalfa leaf with the aim of detecting leaves containing disease. The reported accuracy was impressive, reaching 97.64%. The use of segmentation introduces a more time-consuming process for data annotation when compared to object detection or image classification.

Phadikar et al. [11] researched rice leaf disease identification through the implementation of pattern recognition techniques. The authors proposed the segmentation of the images via an entropy-based bipolar threshold mechanism, post-enhancement of pixel brightness and contrast level. The margins of the disease segment were identified using the eight-connectivity method before being used as inputs to a neural network for the final disease classification. Although the development was based on four datasets, the reported accuracy was below par at 82%.

Rahman et al. [12] focused on the identification of rice leaf diseases through the implementation of convolutional neural networks. The authors proposed a staked-CNN architecture utilizing a two-phase training mechanism for achieving significant reductions in network size whilst retaining respectable classification performance. Comparing their proposed architecture against the VGG-16 network, the authors presented their work as superior, achieving a test accuracy of 95% with a reduction of 98% in terms of network size. 

Huang et al. [13] focused on the detection of leaf blotch disease via UAV image data. The authors proposed the utilization of the stochastic gradient descent (SGD) optimizer for facilitating the backpropagation process during training on the state-of-the-art image classification architecture, GoogleNet. Furthermore, authors aimed to suppress the issue of overfitting via dual data enhancement mechanisms: first, the random discarding of one-band images; secondly, random-based panning of average spectral image brightness. The overall accuracy of the proposed framework on the test data stood at a respectable 92%.

Islam et al. [14] proposed a faster mechanism for rice-leaf-based disease detection via image processing of the affected region. The authors claimed to achieve this by extracting only affected region RGB pixels. The extracted RGB pixel percentages were then used as inputs to a naïve Bayes classifier for classification of the diseases based on one of three categories: rice leaf blast, brown spot, or bacterial leaf blight. The overall accuracy of the classifier post-extraction of the affected regional pixel percentages stood at 89%. 

Asfarian et al. [15] based their research on leaf disease detection via computer vison for integration into pest management. The authors presented a novel mechanism for textural analysis of four types of rice leaf diseases (tungro virus, brown spot, blast, and bacterial leaf blight) by implementing the fractional Fourier. The process involved the conversion of the rice leaf image to a CIELAB color space. The authors reported an overall classification accuracy of 92.5%.

Albattah et al. [16] proposed an AI-based drone system for plant disease detection on the basis of an improved convolutional neural network. Going into the inner details, the authors demonstrated the modifications made to the existing EfficientNetV2-B4 as the addition of dense layers placed at the end of the existing architectural layers. The additional layers assisted with the deep key point classification. The proposed architecture provided an impressive overall performance, achieving an average precision of 99.63%, recall of 99.93%, and accuracy of 99.99%.

Atila et al. [17] also based their research for plant leaf diseases classification on the EfficientNet architecture. Rather than performing any significant modifications to the existing architecture, the authors subscribed to the transfer learning approach. The authors based their training on a large dataset consisting of 55,448 images belonging to 39 classes. The two variants of the EfficientNet tested against other architectures were the B4 and B5 networks. The authors reported highly impressive results for both B4 and B5 given the large number of classes, achieving 99.91% and 99.97% respectively.

Shah et al. [18] focused their research on automated plant diseases classification by presenting a deep learning framework named Residual teacher/student network. The approach was based on the implementation of a CNN architecture for computing the features of the input image. Subsequently, two classification architectures, ResTeacher and ResStudent, plus a decoder mechanism were implemented to enable the final categorization of the abnormalities in the input image. The proposed methodology provided impressive performance with an F1-score of 99.10%. However, in the case of high light-intensity, the architecture was not able to retain its high classification performance. This issue could be addressed or at least mitigated through representative data augmentations focusing on the modeling of on-ground light intensity variations that may occur.

Le et al. [19] proposed a novel framework for detecting crops and weeds that are morphologically similar based on a combination of contour-based masks and filtered local binary pattern operators. The authors claimed the approach improved plant disease recognition capacity by achieving an accuracy of 98.63%; however, the model was not able to achieve respectable performance when facing distortions in the input image. Similar to [18], the use of representative augmentations can be trialed to improve the model’s performance on distorted samples.

Tm et al. [20] proposed a deep learning framework for crop disease categorization. The authors initiated their work with the rescaling of images that were suspected of containing disease before carrying out further preprocessing. Post-processing of the image data, the LeNet architecture was employed for the extraction of representative key points followed by the categorization of the input image as healthy or unhealthy. Due to the lightweight nature of the LeNet, the approach was seen as computationally efficient with a classification accuracy of 95%; however, a significant performance drop was observed with images containing an element of noise.

Summing the literature, we find that although various existing CNN architectures are implemented for leaf disease detection, there is a lack of research based on the development of custom architectures that are tailored toward lightweight application with respect to internal architectural complexity, making them suitable for deployment within a constrained environment.

Furthermore, we observe from various works [18,19,20] that there is a lack of domain-specific augmentations. As a result, the trained models provide highly impressive results on the validation dataset, but when introduced to new test data containing distortions, occlusions, and other forms of variance found in the real-world operating environment, the model performance reduces significantly. Hence, it is important to first comprehend and appreciate the causes of variance for the specific application and then model this variance into the data augmentation strategy. By doing so, not only can the data points be increased, addressing the issue of data scarcity, but also the introduced augmentations would essentially be manifestations of the real variance found in the application environments, hence providing a more robust and generalized model.

### 1.2. Paper Contribution

The contribution of this paper is based on addressing the issues highlighted in the summary of the literature section. The first contribution is based on the representative modeling of variance caused by internal and external factors during the process of rice leaf cultivation. In order to achieve this, various regional surface enhancement augmentations are proposed, tailored toward manifesting variances that may occur in the production environment, i.e., farmland. The proposed augmentations not only result in an increased dataset for addressing the issue of data scarcity but, more importantly, the generated images introduce representative variance and assist with the further accentuating of defective features, resulting in a more generalized architecture.

The first contribution provides a nice segue into the second research objective. That is, the modeling of representative data augmentations facilitates the development of a custom CNN architecture that uses only two convolutional blocks for the classification of input images as containing leaf smut or leaf blight. Furthermore, a reference protocol mechanism is introduced, enabling the suppression of the number of filters used within each block by referencing against the overall number of learnable parameters for state-of-the-art image classification architectures. In doing so, particular care was taken to make sure that the developed CNN architecture was computationally lightweight when compared to other architectures such as the popular ResNet and VGG.

Figure 1 presents a high-level overview of the proposed solution against the conventional approach for leaf disease detection based on transfer learning.

## 2. Methodology

### 2.1. Dataset

This research was based on the detection of two types of rice leaf diseases: blight and smut. Although there are other classes that can also be investigated, the two selected classes provide both ends of the spectrum in terms of their impact on the rice leaf. Rice leaf smut is categorized as a fungus known as Entyloma oryzae. The dataset was created via an internet ‘search and extract’ copyright-free process using Google image search. The images consisted of varying resolution; as a preprocessing step, all images were capped at a resolution of 224 × 224 pixels. The main visual characteristic of this disease is the emergence of black spots on the leaves. Although leaf smut by itself does not cause any significant damage to the rice leaves, it can increase the vulnerability of the rice leaves to other diseases. Figure 2a presents a rice leaf containing leaf smut.

Bacterial blight, on the other hand, is categorized as one of the more deadly diseases that can have a destructive impact on the cultivation of rice leaves. With regard to its emergence, bacterial blight is initiated as water-soaked streaks spread from the margins and leaf tips of the leaf, becoming larger in size, and releasing a milky-looking ooze that consolidates into yellowish droplets. During its progression, gray-whitish lesions emerge on the leaf surface, signifying the final stage, as the leaf dries and eventually dies. Figure 2b presents a sample image of a rice leaf containing bacterial blight.

The original dataset is presented in Table 1. The fundamental issue with the original dataset was its small size. It is a well-known fact that CNNs require large amounts of training data in order to truly generalize the underlying features. However, in this case, as evident from the comparison between the two classes in Figure 2, the two classes had a noticeable visual difference that could help to differentiate between the two types. That is to say, the smut class usually manifested brown spots whilst the more severe case of blight was represented via yellowish regions across the leaf surface. However, both causes were random in size and non-uniform in their locality on the leaf surface. Hence, modeling of the apparent variations was performed through selected image processing to obtain a scaled and representative dataset. Though the scaled dataset would not be drastically larger in size compared to the original dataset, the rationale for suppressing the size of the dataset post-augmenting was due to the fact that it would be trained on a custom lightweight architecture, where we would want the internal parameters to learn the true variance rather than overfit due to the large number of repetitive samples. 

### 2.2. Proposed Methodology for Representative Data Scaling

Two fundamental motivations led to the development of a specific data transformation mechanism presented as ‘Domain Feature Mapping (DFM)’, shown in Figure 3: first, the lack of data samples within the original dataset (40 samples for each class); secondly, due to the internal variance and similarity between the two classes, as discussed earlier regarding the differentiation features between the two classes, i.e., brown spots and yellowish regions. However, the surface of the leaf, i.e., the background on which the two signs occur, was similar. Hence, in the cases of the blight, i.e., yellowish regions appearing small in size may be misclassified as smut and, hence, not given much importance, as smut is a minor disease with no significant implication compared to blight. This misclassification could have a detrimental impact on the cultivation process, if the trained model is simply generalized on the false assumption that all small-sized yellowish regions imply leaf smut. Each processing technique presented in the DFM phase is discussed in detail in the following sections.

Figure 3 presents an overview of the proposed methodology, starting with the modeling of variance and data scaling via DFM to the development of the custom CNN. The development of the internal convolutional blocks of the custom CNN was assisted via the introduced Reference Protocol (RP) mechanism. This essentially assured that the developed architecture was lightweight in terms of its computational learnable parameters by benchmarking against the learnable parameters of various state-of-the-art image classification architectures. The emphasis on developing an architecturally and computationally lightweight architecture was so that it could be deployed at scale within the quality inspection process for rice leaf cultivation without requiring specific GPU-enabled hardware.

### 2.3. Class Accentuation

The first phase of the DFM, named class accentuation, aimed at essentially enhancing the contrast between defective pixels and the background. This was due to potential misclassification that may be caused by the arbitrary size and non-uniform distribution of the two classes internal characteristics. The class accentuation process was initiated by transforming the input image into greyscale for mitigating the background color interference. Figure 4 presents the transformation processes for a given rice leaf containing blight: (A) original image; (B) grey-scale conversion. 

It can be seen from the conversion in Figure 4B that although the blight is clearly visible on a significant portion of the surface, the background contrast does not relay this. Therefore, as part of the class accentuation process, contrast-enhancement via global histogram equalization was implemented.

Figure 5A shows the original image transformed into greyscale whilst (B) presents the greyscale image post-global histogram equalization. It is clear from the output image that global equalization was not an appropriate technique for class accentuation. That is, due to the fact that the background pixels post-equalization had intensified along with the boundary of the leaf, the introduction of noise at the margins was also observed. The explanation for this was simply based on the nature of the equalization. As the equalization was globally applied across the image, the background had also been accentuated; hence, rather than increasing the defective region contrast, a general, indiscriminate increase across all pixels was observed.

To address the issues highlighted with global equalization, regional equalization based on pre-defined pixels as equalization regions was implemented. This was performed via the Contrast-Limited Adaptive Histogram Equalization (CLAHE) technique. Figure 6 shows the comparison between (A) Global equalization and (B) CLAHE for blight and (C) CLAHE for smut. It can be observed that the latter was successful in addressing the issue of over-amplification caused by the former. Furthermore, the element of noise introduced at the margins of the leaf had also been addressed via CLAHE. The regional space, i.e., each equalization tile, was set to 8 by 8 pixels. The contrast enhancement between the background and the blight region had clearly increased post-CLAHE as compared to that of global equalization. This would assist the CNN during the training phase with better generalization capacity as it would require less convergence efforts by the optimizer for differentiating between the background and defective regions.

### 2.4. Device-Based Translation

The second DFM technique was aimed at generating variance that may occur due to the hardware utilized. The aim of the research was to provide a lightweight architecture that could be deployed on computationally constrained hardware devices rather than requiring cloud-based inference. The rationale for this was due to the fact that rice leaf cultivation is carried out mostly in hot Middle Eastern countries, which may not necessarily have the communications infrastructure to facilitate cloud AI inference. Producers that do have the capacity in terms of the communications infrastructure may still be reluctant due to cyber security issues with their business-related data being processed in a remote location. 

Figure 7 presents two potential hardware options for deploying and inferencing based on the developed CNN architecture presented in the subsequent sections. Using either the jetson nano-coupled with a drone (Figure 7A) or a Raspberry Pi (Figure 7B) attached to the production line would generate new variance in terms of image orientations with respect to their placement configurations, which would vary based on the production site.

For example, a Raspberry-Pi-based system would carry out inference by processing the images via the camera and running the onboard CNN model. The rice leaf plants running down the production line can come in various orientations. Hence, if the model is not trained on a dataset containing images at various orientations, it may falsely assume that they belong to a particular class. Figure 7C shows the variations within the orientations and aspect ratios that could result from the varying configurations/placement of the hardware capturing device. 

After realizing the importance and impact hardware-induced variance can have on the model’s performance, an induced shift mechanism was implemented for generating images with a pre-defined pixel shift parameter. Thus, specifying the shift parameter in the form of ($h_{x},h_{y}$), the transformation matrix is presented as:(1)$$M=\left[\begin{array}{ccc} 1 & 0 & h_{x}\\ 0 & 1 & h_{y} \end{array}\right]$$

The matrix ‘*M*’ would be converted to an array before applying an affine transform, where *inp* = input image, *otp* = output image, and *M* = transformation matrix:(2)$$otp\left(x,y\right)=inp(M_{11}x+M_{12}x+M_{13},M_{21}x+M_{22}y+M_{23}$$

The generated images, as a result of pixel-shifting, captured variance caused by the aspect ratio and hardware placement, resulting in losing random marginal-based pixel data. However, this transformation also resulted in some cases of excluding pixels containing the defective regions. As mentioned earlier, the defects are indiscriminate in their locality selection with respect to the surface of the rice leaf plant. Hence, to introduce additional variations by preserving the defective regions that may occur at the margins of the image, center-based rotations were introduced. Traditionally, the rotation of an image with respect to an angle (*ϴ*) is attained via the matrix:(3)$$Mahf=\left[\begin{array}{cc} cosϴ & -sinϴ\\ sinϴ & cosϴ \end{array}\right]$$

Yet, our objective is on anchoring the center of the image for the rotation; hence, the matrix becomes
(4)$$Mahf=\left[\frac{\alpha \beta \left(1-\alpha \right)\cdot centre.x-\beta \cdot centre.y}{-\beta \alpha \cdot centre.x+\left(1-\alpha \right)\cdot centre.y}\right]$$
where *centre* = rotation center (input image), *ϴ* = rotation angle (degrees), and *scale* = isotropic scale factor.
(5)α=scale·cosϴ,  β=scale·sinϴ

### 2.5. Bilateral Filtering

The final technique proposed in the DFM for scaling the original dataset in a representative manner was bilateral filtering. It was observed during data inspection that both defects smut and blight contain boundaries around the defective regions. Although these may be fuzzy in certain images, it would be these boundaries coupled with the color contrast that would enable the differentiation of the defects from the background surface, as shown in Figure 8.

Brown spots (smut) and yellowish regions (blight) appearing on the leaf surface are non-uniform in their shape but do consist of margins, as shown in Figure 8. These margins subject to the level of contrast with respect to the leaf surface background can appear distinctive or blurred. Thus, the modeling objective in this case was to effectively reduce leaf surface (background) noise whilst conserving the starkness of margins around the defective regions. The initial component of the bilateral filter was fundamentally linear Gaussian smoothing:(6)$$g\left(x\right)=\left(f*G^{s}\right)\left(x\right)=\int_{R}f\left(y\right)G^{s}\left(x-y\right)dy$$
where f(y) weight = $G^{s}\left(x-y\right)$ depending solely on the spatial distance of ‖x−y‖.

Bilateral filtering presents a weighting component dependent on the tonal distance of f(y)−f(x); hence, the equation becomes
(7)$$g\left(x\right)=\frac{\int_{R}f\left(y\right)G^{s}\left(x-y\right)dy\;G^{t}\left(f\left(x\right)-f\left(y\right)\right)dy}{\int_{R}G^{s}\left(x-y\right)\;G^{t}\left(f\left(x\right)-f\left(y\right)\right)dy}$$

Notice that the weights depend only on the image values; hence, normalization is necessary to make sure the sum of all weights is equal to one.

Figure 9 presents a sample set of the rice leaf containing both smut and blight post-bilateral filtering. The blurring of the leaf surface (background) whilst maintaining the defect margins resulted in the enhancement of defective regional margins with respect to the background surface.

### 2.6. Proposed Architecture Design Mechanism

Table 2 presents the scaled dataset based on the augmentations applied via the proposed DFM after implementing data splitting for training validation and testing sets with a split ratio of 70%, 15%, and 15%, respectively.

This section of the research focuses on the development of the lightweight CNN architecture for the classification of smut and blight rice leaves. As mentioned earlier, rice cultivation is usually undertaken in hot countries, especially in the Middle East/Asia continent. Countries that come under the category of developing nations may not necessarily have the infrastructure for facilitating automated quality inspection process, especially in the rural areas. This was the inspiration for developing a custom lightweight architecture rather than subscribing to the transfer learning approach via state-of-the-art architectures. By doing so, rice producers could utilize the automated inspection system on a constrained device such as a Raspberry Pi rather than requiring cloud connectivity for hosting computationally demanding architectures. 

Figure 10 presents the proposed framework for the development of the internal convolutional and fully connected layers. The defining of these two components was critical as it had a direct impact of the computational composition of the resultant architecture. That is to say, the convolutional block consisted of filters, and the weights of these filters would be learned during the training phase and then frozen for test inferences. Hence, the greater the number of convolutional filters, the greater the increase in the overall computational parameters. Similarly, the fully connected layers consisted of densely populated neurons, the weights of which would also be learned during training and then frozen for test and deployment. Therefore, to assist with the development of the number of internal layers, we present a Reference Protocol (RP) Mechanism. This essentially provided the learnable parameters of various state-of-the-art architectures. 

The process was initiated by developing a two-layered CNN, i.e., containing two convolutional blocks and two fully connected layers. The parameters of the model were calculated based on the number of filters used within each convolutional block and fully connected layers. The total number of parameters Gz were compared with the parameters for the models within the RP block. If the parameters were greater than those for any of the architectures in the RP, then the number of filters within the convolutional blocks would be reduced until the developed architecture was smaller in computational parameters as compared to those architectures in the RP. Once this was achieved, the developed architecture was ready for training on the dataset presented earlier in Table 2. However, if the training stage provided low performance due to the insufficient capacity of the developed architecture, then the proposed work would not have any significance as further increasing the computational parameters above the RP architectures would imply that the developed architecture is no longer lightweight. 

Looking deeper at the convolutional blocks, each block consisted of a number of filters (learnable parameters), an activation function, and feature map aggregation. Max-pooling was selected for resultant feature map aggregation with the aim of mitigating the impact of positional dependency as the defects were indiscriminate in their locality selection.
(8)$$u_{xy}^{l}=max_{i=0,\ldots ,s,j=0,...,s}u_{\left(x+i\right)\left(y+j\right)}^{l}$$
where $u^{l}$ = activation of layer l.

We noted that several activation functions have been utilized in the development of deep learning architectures. The rationale for selecting ReLu as the activation function for our CNN was due to its effectiveness in combatting the vanishing gradient issue. This was a major limitation for the sigmoid and tanh activation functions. Furthermore, diving deeper into the mathematical structure of the ReLu, we observe that it is simply a ‘Max’ operation and, hence, computationally more lightweight.
(9)z(x)=max(0,x)

Additionally, batch normalization was introduced within the two convolutional blocks. The rationale was to overcome the internal covariate shift, which may be caused due to the internal class variance, impacting the performance of the architecture in terms of convergence speed. For example, the severity of the smut class, i.e., number of brown spots and their intensity, would differ from image to image, although they belong to the same class. In the case of no batch normalization, this internal variance would result in the projection of the two images onto a different feature space, requiring more training effort (time) for the model to converge.
(10)$$B_{n}\left(z_{i}\right)=\gamma_{i}{\hat{z}}_{i}+\beta_{i}$$
(11)$${\hat{z}}_{i}=\frac{z_{i}-E\left[z_{i}\right]}{\sqrt{var\left[z_{i}\right]}}$$
where $z_{i}$ = pre-activation, *γ*, *β* = Learnable parameters.

The developed CNN architecture and the resultant computational parameters are presented in Figure 11. A comparison of the RP architectures against the developed architecture is presented in the results section. The number of learnable parameters for each internal block layer were calculated via:(12)$$\;\left(B*M*f_{in}+1\right)*f_{out}$$
where B, M = filter dimensions, $f_{in}$ = number of input filters, and $f_{out}$ = resultant output feature maps.

## 3. Results

### 3.1. Hyper-Parameters

This section of the article presents the performance of the trained CNN across various metrics. In order to train the CNN, a set of hyperparameters had to be defined. The hyper-parameters used for training on the dataset are provided in Table 3.

### 3.2. Reference Protocol (RP) Comparison

Before undergoing the training of the CNN, as per the framework presented earlier in Figure 10, the first criterion the proposed CNN would be required to meet was the number of learnable parameters being smaller than any of the architectures mentioned in the RP mechanism. Table 4 presents the computational comparison between the developed CNN architecture and that of the state-of-the-art image classification architectures selected in the RP.

It can be observed that our proposed CNN architecture was significantly smaller in its computational parameters, compared to all RP architectures. ResNet-18 was the second most effective model in terms of learnable parameters with a difference of 1.76 million parameters. VGG-19 consisted of a significantly higher number of learnable parameters (143.67 million) compared to all other architectures. This demonstrates the reason why the VGG-19 architecture is not deployed onto edge devices, as its computational demand is too high, requiring specific GPU hardware for reasonable inference performance post-deployment.

### 3.3. CNN Training and Validation

After validating the lightweight architecture of our proposed CNN, the next stage was the training and validation of the architecture. The architecture was trained on the training dataset and validated on the validation set (Table 2) as per the hyperparameters mentioned in Table 3.

Figure 12 presents the training and validation graph showing the progression of the CNN training. It can be seen that the architecture was able to achieve a training accuracy of 90.1% with an impressive validation accuracy of 92.9%. There are a couple of points to highlight here that indicate the successful data inspection and proposed CNN development framework.

First, the fact that the difference between the training and validation accuracies is small (2.8%) signifies that the proposed augmentations were successful in generating a representative sample for the model to generalize without overfitting. Secondly, the size of the dataset post-augmentations was still small in conventional terms (total samples = 560). This demonstrates that it was not the increased size of the dataset but rather the effectiveness of the augmentations and selection of internal CNN layers that contributed toward the high performance of the architecture. 

Further inspection of the internal class performance via the confusion matrix is shown in Figure 13. As evident from the confusion matrix, both classes were successful in correctly predicting the true class of the images.

### 3.4. Test Data Performance

After achieving high performance on the training and validation data, the final performance check was based on the introduction of test data. In general, the objective is to hide the test data throughout the training and validation process in order to mitigate the risk of data leakage. Once the final architecture has been trained and the validation performance is sufficient, the test data can then be introduced to validate if the model had in fact truly generalized.

In order to achieve this, first, the training and validation datasets were merged into one folder known as the training dataset. The architecture would be retrained on the merged training dataset based on the same hyperparameters outlined in Table 3. The trained model would then be validated on the actual test dataset. In order to provide granular performance across various metrics, the performance was benchmarked with respect to the precision, recall F1-score, and the accuracy, as shown in Table 5.

Starting with the accuracy, it can be stated that the accuracy on the test data had actually improved from the accuracy presented in the validation stage by a margin of 1.2%. This demonstrates the high generalizability of the developed architecture. Furthermore, we observe that all metrics are in close conformity with respect to their results. This again is a manifestation of the successful data augmentation strategy for generating representative data samples and also the proposed CNN development framework for developing a lightweight yet highly performant architecture.

## 4. Discussion

The inception of this research was with the investigation of the limited dataset for understanding the practical variance that may occur within the cultivation process of rice leaf both due to internal and external factors. As a result, various augmentation techniques were proposed under the Domain Feature Mapping (DFM) mechanism. The aim of DFM was not to simply increase the dataset in size but rather to concentrate on producing representative samples to facilitate the proposed CNN toward true generalization. This is evident from the fact that, post-DFM, the dataset had increased from 80 to 560 samples as opposed to acquiring thousands of images via applying of arbitrary augmentations. We feel that the proposed DFM process will provide researchers focusing on automation in the rice cultivation domain with a set of techniques that can output representative samples, helping them to address the issue of data scarcity without compromising on the quality of the data.

Moving onto the proposed CNN development framework, it can be said that based on the performance of the trained architecture on the test dataset, the CNN, although containing only 2 convolutional blocks with limited filters, was able to highly generalize on the dataset. The proposed RP mechanism and its assistance in developing a lightweight architecture consisting of a limited number of computational parameters can again be utilized by developers for designing architectures that are computationally friendly and can be deployed without requiring specific GPU hardware. 

It is also important to mention that the constrained environment for undertaking this research is also a manifestation of the high efficacy of the proposed mechanisms. For example, the development and training of the CNN were carried out in the Google Colab environment. Due to the limited availability of computational resources for the free tier, the training was limited to 125 epochs, whereas, conventionally, CNN architectures are trained for several hundreds or thousands of epochs for achieving acceptable performance. Based on the training of the proposed CNN for 125 epochs with a limited dataset in terms of size, we achieved an F1-score of 94.0% on the test data.

## 5. Conclusions

In summary, we review our research output with respect to the research objective. The objective was the development of a lightweight deep learning architecture that could be deployed in constrained environments for automated detection of leaf smut and leaf blight, assisting with the quality inspection process in rice leaf cultivation. 

In order to achieve this objective, the first barrier we had to overcome was the data scarcity issue with the original dataset consisting of 80 images. We considered this as a practical issue that other developers may face due to the difficulty in procuring certain datasets based on the application. Hence, we provided a series of processing techniques under DFM. The proposed techniques were a result of careful domain modeling based on variances that may occur from external and internal factors.

Furthermore, the CNN development framework utilizing the RP as the filter reference protocol enabled the development of a CNN architecture consisting of only 9.43 million learnable parameters compared to 143.67 million for VGG-19. Coupling the two proposed frameworks together, we were able achieve a highly accurate architecture with an overall test accuracy 94.1%

We feel that our proposed methodology will enable developers in the field of CNN-based quality inspection and fault detection via AI in general [21,22] across various domains including industrial [23], medical [24], and renewable energy [25] to address the common issues that may occur in specialist applications due to data scarcity, variance modeling, and lightweight internal filter configurations. 

## Figures and Tables

**Figure 1 foods-11-03914-f001:**
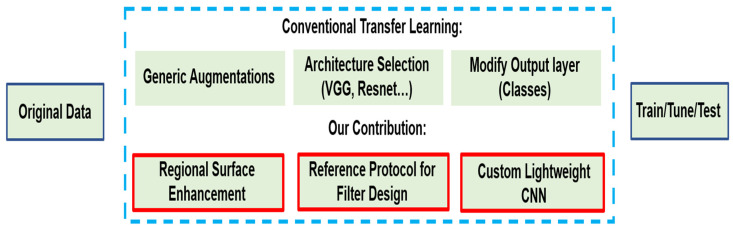
Abstract Solution Comparison.

**Figure 2 foods-11-03914-f002:**
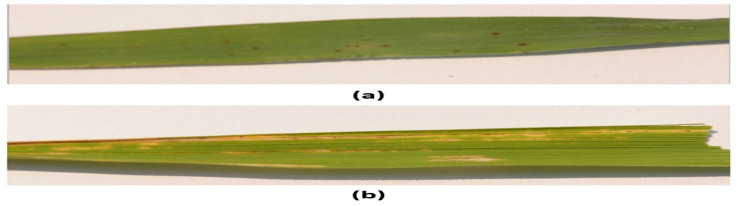
Class Samples: (**a**) leaf smut; (**b**) bacterial blight.

**Figure 3 foods-11-03914-f003:**
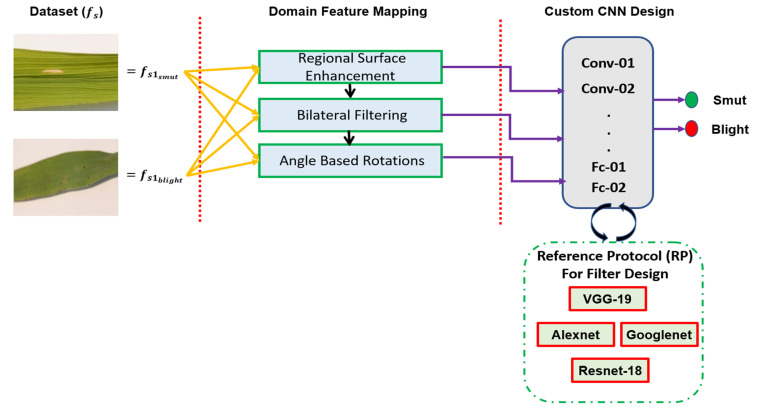
Methodology Overview.

**Figure 4 foods-11-03914-f004:**
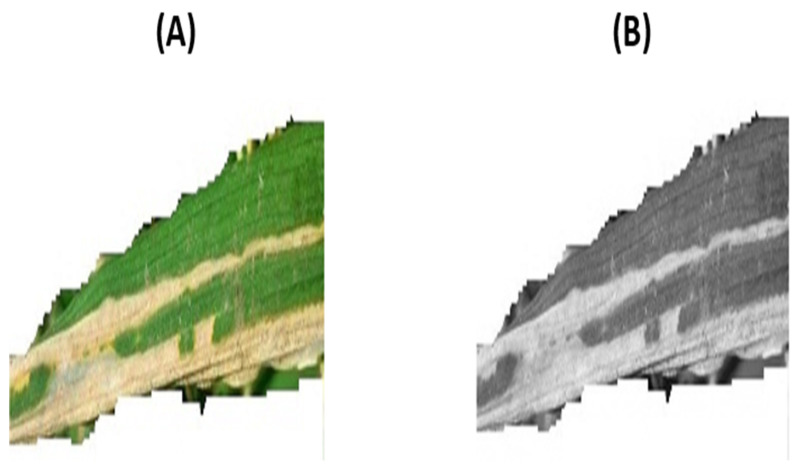
Image Conversion and Histogram (**A**) original image (**B**) grey-scale.

**Figure 5 foods-11-03914-f005:**
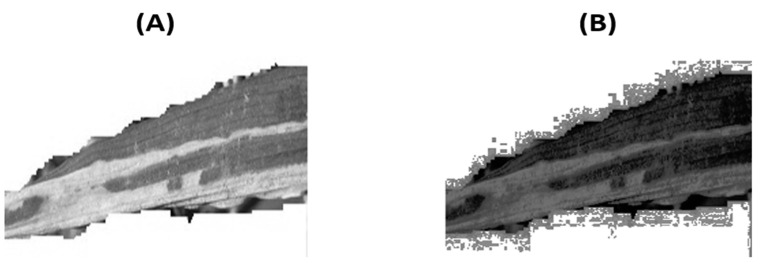
Global Equalization: (**A**) original greyscale; (**B**) post-equalization.

**Figure 6 foods-11-03914-f006:**
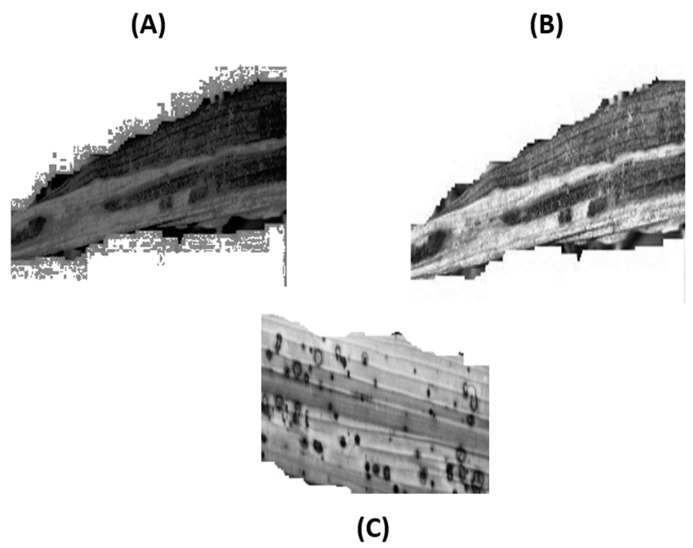
Equalization comparison: (**A**) Global; (**B**) CLAHE for blight; (**C**) CLAHE for smut.

**Figure 7 foods-11-03914-f007:**
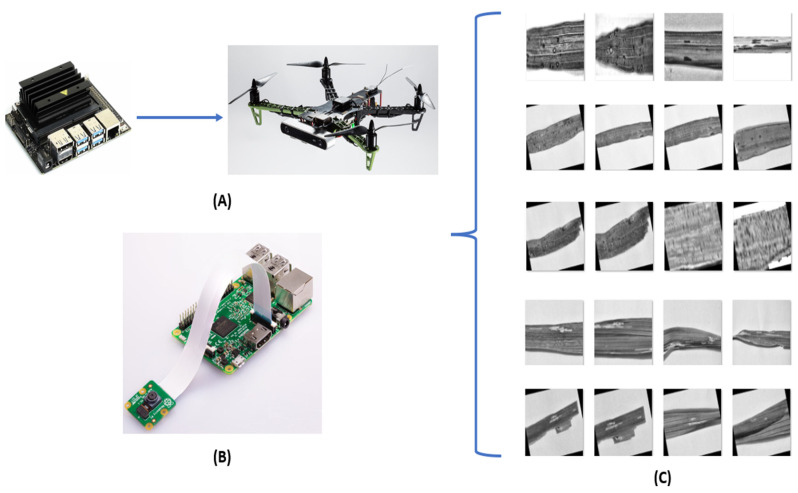
Fundus Procurement Process.

**Figure 8 foods-11-03914-f008:**
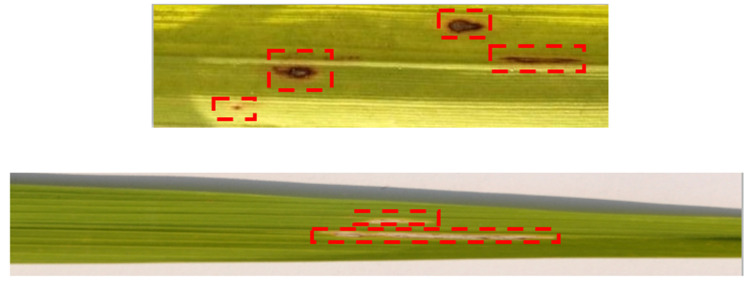
Observing Margins for defective regions.

**Figure 9 foods-11-03914-f009:**
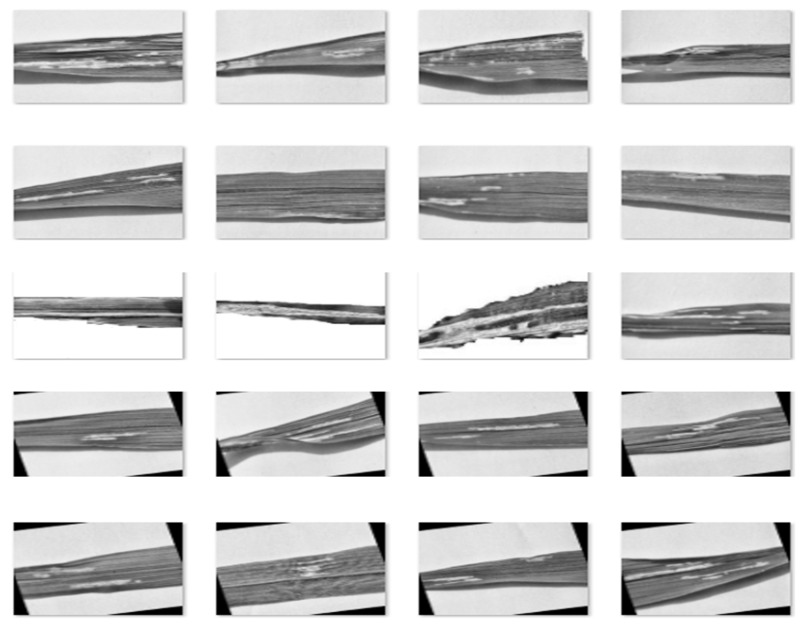
Generated Augmentations For Bilateral Filtering.

**Figure 10 foods-11-03914-f010:**
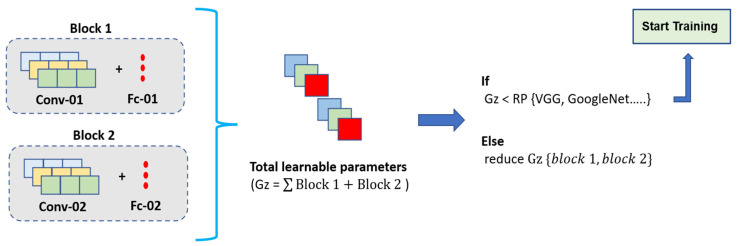
Proposed Framework For CNN Development.

**Figure 11 foods-11-03914-f011:**
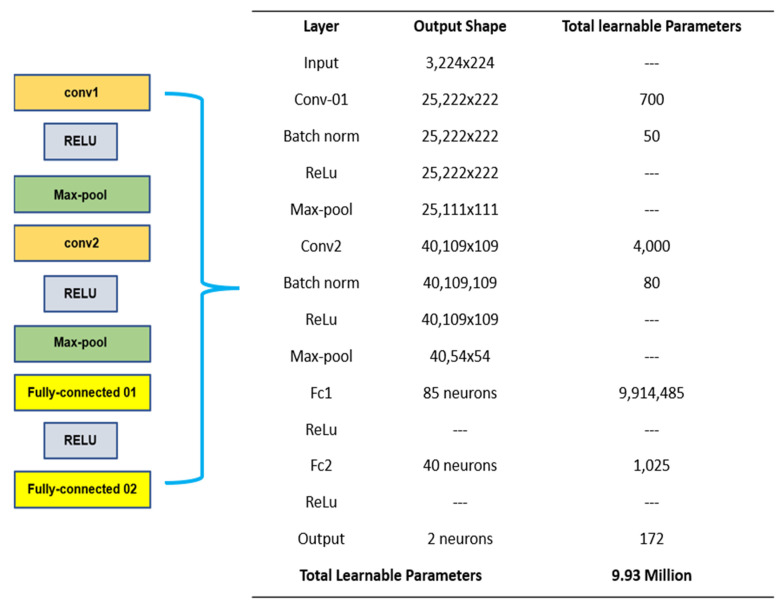
Proposed CNN Internal Block Diagram.

**Figure 12 foods-11-03914-f012:**
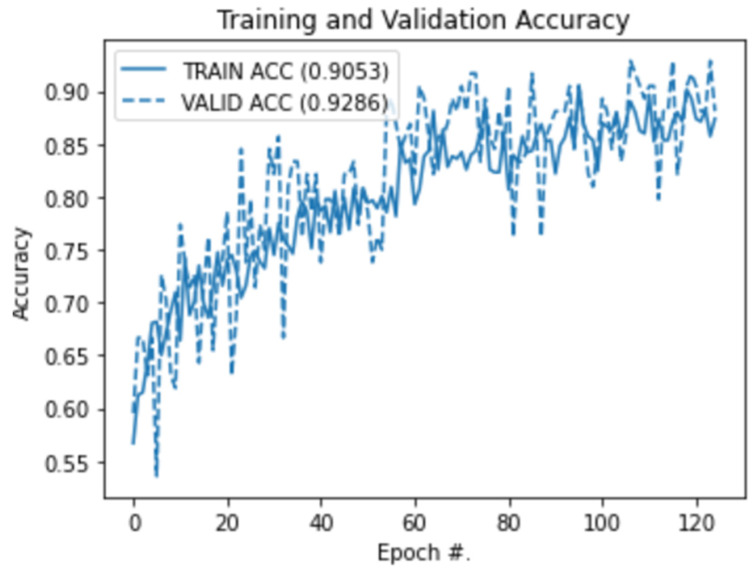
Training and Validation Performance.

**Figure 13 foods-11-03914-f013:**
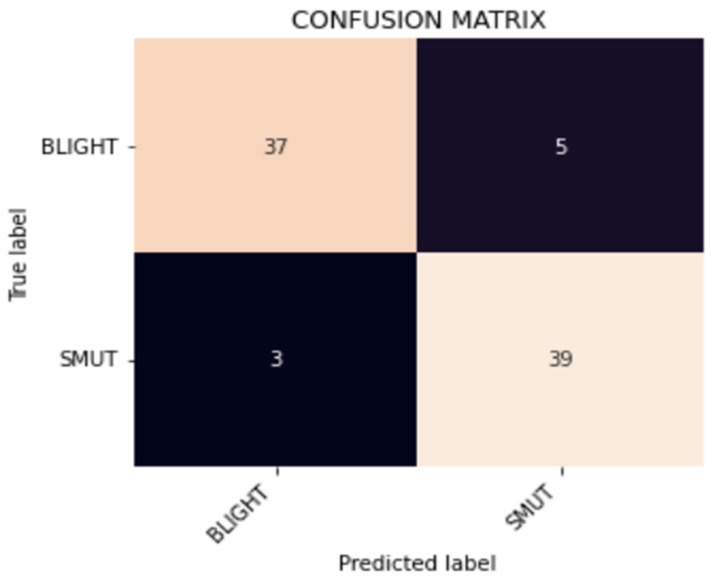
Confusion Matrix For Trained Model.

**Table 1 foods-11-03914-t001:** Original Dataset.

Smut	40
Blight	40

**Table 2 foods-11-03914-t002:** Training Dataset.

Class	Training	Validation	Test
smut	196	42	42
blight	196	42	42

**Table 3 foods-11-03914-t003:** Hyperparameters.

Batch Size	4
Epochs	125
Learning Rate	0.001
Loss	Cross Entropy
Optimizer	SGD-M

**Table 4 foods-11-03914-t004:** Reference Protocol (RP) Comparison.

Architecture	Parameters (M)
Our Model	9.93
VGG-19	143.67
ResNet-18	11.69
AlexNet	61.0
Googlenet	13.0

**Table 5 foods-11-03914-t005:** Test Set Performance.

Metric	Score (%)
Accuracy	94.10
Precision	95.0
Recall	93.0
F1-score	94.0

## Data Availability

The data are available from the corresponding author.

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
