# Peer review of "Feature Mapping for Rice Leaf Defect Detection Based on a Custom Convolutional Architecture"

_foods, 2022, doi:10.3390/foods11233914_

Round 1
Reviewer 1 Report
1. The authors should pay attention to small and big letters, such as in line 56, "reduction. it then addresses the issue ...", etc.
2. In line 54, "by via a two ..."?
3. In line 60, "accuracies for all defective categories our impressive,"?
4. In line 62, "hence we focus firstly on". Who are "we" here?
5. In line 68, "Hardware" should be hardeware. Typos will not be pointed out one by one for the rest of the manuscript. The authors should check their writing carefully and seriously to eliminate them.
6. In line 69, "F Qin et al"
7. In line 72, "however the use of segmentation but introduced more load at the data annotation phase and also result in computationally more demanding architectures." Wrong sentence!
8. In line 76, "The propose the segmentation of the image via the entropy-based bipolar threshold mechanism post enhancement of pixel brightness and contrast level." Wrong sentence! There are more gramatically wrong sentences than this reviewer can pick them out one by one in the manuscript. These wrong sentences refrain the reader from understanding the paper.
9. Be careful about using English tenses: present tense and past tense. The tense should be consistent when describing other researchers' work.
10. In line 139, "Tm et al [20]"?
11. It seems the authors do not know well the guidlines for scientific writing.
12. Is such a simple table as Table 1 necessary?
13. What are the horizontal and vertical variables in Fig. 4(C)?
14. It is not understandable from line 350 to line 359.
15. The authors use combination of common AI techniques for identifying two types of rice leaf diseases. Actually, there are many AI based methods which can identify more types of plant leaves in the literature. Therefore, no sufficient novelty is shown in this paper.
16. This paper proposes a lightweight model, but there is no comparision of their model with other normal or "heavyweight" models. By looking into the confusion matrix in Fig. 13, this reviewer feels that the predicting is not accurate enough.
Author Response
Dear Reviewer,
Point raised:
‘The authors should pay attention to small and big letters, such as in line 56, "reduction. it then addresses the issue ...", etc.’
Answer:
Firstly, thank you for the positive and constructive feedback. This has been addressed after proof-reading.
Point raised:
‘ In line 54, "by via a two ..."?’
Answer:
This has been addressed after proof-reading.
Point raised:
‘In line 60, "accuracies for all defective categories our impressive,"?.’
Answer:
This has been addressed.
Point raised:
‘In line 62, "hence we focus firstly on". Who are "we" here?’
Answer:
This has been addressed.
Point raised:
‘In line 68, "Hardware" should be hardware.’
Answer:
This has been addressed.
Point raised:
‘In line 69, "F Qin et al’
Answer:
This has been addressed throughout the literature review.
Point raised:
‘In line 72, "however the use of segmentation but introduced more load at the data annotation phase and also result in computationally more demanding architectures." Wrong sentence!’
Answer:
The sentence has been rewritten in the correct manner.
Point raised:
‘In line 76, "The propose the segmentation of the image via the entropy-based bipolar threshold mechanism post enhancement of pixel brightness and contrast level." Wrong sentence!’
Answer:
This has also been written again in the correct manner.
Point raised:
‘Be careful about using English tenses: present tense and past tense. The tense should be consistent when describing other researchers' work.’
Answer:
This has been addressed.
Point raised:
‘In line 139, "Tm et al [20]"?’
Answer:
This has been corrected throughout the literature review.
Point raised:
‘Is such a simple table as Table 1 necessary?’
Answer:
Table 1 presents the initial dataset, demonstrating the lack of images in a tabular manner. This is followed by Table 2 showing the dataset post processing, augmenting and splitting. This is a theme observed across various papers, in particular those focusing on representative data scaling.
Point raised:
‘What are the horizontal and vertical variables in Fig. 4(C)?’
Answer:
Figure 4 (C), was not required here and hence has been removed as per the text description.
Point raised:
‘It is not understandable from line 350 to line 359.’
Answer:
This is presenting the mathematical composition for bilateral filtering. Furthermore, all equations have been numbered now.
Point raised:
‘The authors use combination of common AI techniques for identifying two types of rice leaf diseases. Actually, there are many AI based methods which can identify more types of plant leaves in the literature. Therefore, no sufficient novelty is shown in this paper.’
Answer:
The DFM is essentially the grouping of various image pre-processing techniques for representative data generation. Additionally, the Reference Protocol (RP) proposed for the custom architecture development is a novel contribution for determining the number of layers within the internal blocks of the architecture. the effectiveness of this is demonstrated through comparison with other architectures with respect to the number of internal parameters. Please refer to section 2.6.
Point raised:
‘This paper proposes a lightweight model, but there is no comparision of their model with other normal or "heavyweight" models. By looking into the confusion matrix in Fig. 13, this reviewer feels that the predicting is not accurate enough.’
Answer:
The comparison was done against various state-of-the-art architectures, in order to prove the effectiveness of the proposed RP with respect to the computational footprint. The reason for benchmarking on the internal parameters was due to the need to demonstrate the effectiveness of the proposed RP in developing a custom yet lightweight architecture. Only after the lightweight footprint was proven (refer to Table 4), other performance evaluation carried out.
Reviewer 2 Report
The paper presents a convolutional neural network for the detection of rice leaf defects. The major novelty of the proposed work is the use of pre-processing techniques for data augmentation and to increase the contrast between images of different groups as well as the use of a lightweight CNN that can be implemented using portable processing systems. The following revisions are needed to improve the paper:
1) The proposed CNN classifies the input images into two different categories, corresponding to two different defects. However, to correctly discriminate between healthy and unhealthy rice leafs, I think the authors should have discriminated the samples in three different categories: healthy, blight defect and smut defect.
2) In section 2.4 the authors propose two different strategies to collect leaf images. However, the authors should discuss in detail how they have obtained the initial 80 images (taken by camera or from a public database) as well as to present the images characteristics (resolution, etc.).
3) The proposed framework to build the CNN layers presented at pages 14-15 is not very clear and must be presented more clearly.
4) Some errors and typos must be corrected: line 55, line 111 and line 169, capital letter after “.”; line 60 “are” and not “our”; line 63 “starting” and not “staring”; line 73 “but” must be deleted and “result” should be “results”; line 76 “They” and not “The”; line 139 the authors of ref [20] are Hussain et al.; line 237 “is discussed in detail in the preceding sections”, but DFM techniques are discussed in following sections.
Author Response
Dear Reviewer,
Point raised:
‘The proposed CNN classifies the input images into two different categories, corresponding to two different defects. However, to correctly discriminate between healthy and unhealthy rice leafs, I think the authors should have discriminated the samples in three different categories: healthy, blight defect and smut defect.’
Answer:
Firstly, thank you for the positive comments and constructive feedback. The aim of this particular paper was to demonstrate the effectiveness of the RP mechanism and effective data generation, as this has been proven from the paper, yes, indeed we can focus on more classes and tune the architecture accordingly.
Point raised:
‘In section 2.4 the authors propose two different strategies to collect leaf images. However, the authors should discuss in detail how they have obtained the initial 80 images (taken by camera or from a public database) as well as to present the images characteristics (resolution, etc.).’
Answer:
The purpose of this section was in fact to demonstrate the variance in images that can be induced due to hardware i.e., placement/ orientation etc and how the proposed DFM can model this effectively. Please refer to the text in 2.4.
Point raised:
‘The proposed framework to build the CNN layers presented at pages 14-15 is not very clear and must be presented more clearly.’
Answer:
The first part of the proposed architecture section looks at the inner details of the proposed RP coupled with details on CNN selection. This is resultant CNN is presented in Figure 11, in order to provide a more detailed process view.
Point raised:
‘Some errors and typos must be corrected: line 55, line 111 and line 169, capital letter after “.”; line 60 “are” and not “our”; line 63 “starting” and not “staring”; line 73 “but” must be deleted and “result” should be “results”; line 76 “They” and not “The”; line 139 the authors of ref [20] are Hussain et al.; line 237 “is discussed in detail in the preceding sections”, but DFM techniques are discussed in following sections.’
Answer:
These have been addressed, accordingly, thank you.
Reviewer 3 Report
The article is interesting and presents an innovative approach in the subject matter analysed.
In my opinion, however, the paper is outside the scope of the journal.
Comments
1. The authors should prepare the article according to the Instructions for Authors. Especially in terms of literature, numbering of mathematical formulas.
2. The abstract should be revised. To a greater extent, it should be a brief summary of the manuscript.
3. In the Discussion section, the authors should contrast their own results with those of other authors.
4. Standardise the font size in the text of the article.
5. Subsection 2.5 is missing in the text. There is 2.4 and 2.6.
Author Response
Dear Reviewer,
Point raised:
‘The authors should prepare the article according to the Instructions for Authors. Especially in terms of literature, numbering of mathematical formulas.’
Answer:
Firstly, thank you for the positive comments and constructive feedback. This has been addressed, in particular all equation have been numbered.
Point raised:
‘The abstract should be revised. To a greater extent, it should be a brief summary of the manuscript.’
Answer:
The aim was to provide a brief premise for the research followed by our contribution i.e., variance modelling and custom CNN development for the particular application. In-depth details are presented in the main sections of the paper.
Point raised:
‘In the Discussion section, the authors should contrast their own results with those of other authors.’
Answer:
This comparison has been presented in the result section as it was required for demonstrating the effectiveness of the architecture with respect to architectural footprint before any further testing could be done.
Point raised:
‘Standardise the font size in the text of the article.’
Answer:
This has been addressed.
Point raised:
‘Subsection 2.5 is missing in the text. There is 2.4 and 2.6.’
Answer:
This has been addressed.
Round 2
Reviewer 2 Report
The authors should make the following revision. In Section 2.1 (Dataset) the authors should explain how they have obtained the 40+40 images of the original dataset (taken by camera or from a public database) as well as to present the images characteristics (resolution, etc.).
Author Response
Reviewer point:
The authors should make the following revision. In Section 2.1 (Dataset) the authors should explain how they have obtained the 40+40 images of the original dataset (taken by camera or from a public database) as well as to present the images characteristics (resolution, etc.).
Answer:
Thank you for your constructive feedback, we have addressed this within the report as follows
'The dataset was created via an internet ‘search and extract’ copyright-free process using Google image search. The images consisted of varying resolution, as a pre-processing step, all images were capped at a resolution of 224x224 pixels. The main visual characteristic of this disease is the emergence of black spots on the leaves. Although leaf smut by itself does not cause any significant damage to the rice leaves, it can increase the vulnerability of the rice leaves to other diseases. Figure 2 (a) presents a rice leaf containing leaf smut.'